# Direct oxygen isotope effect identifies the rate-determining step of electrocatalytic OER at an oxidic surface

Sandra Haschke[1], Michael Mader[2], Stefanie Schlicht[1], André M. Roberts [2], Alfredo M. Angeles-Boza [3], Johannes A.C. Barth[2] & Julien Bachmann [1,4]

Understanding the mechanism of water oxidation to dioxygen represents the bottleneck towards the design of efficient energy storage schemes based on water splitting. The investigation of kinetic isotope effects has long been established for mechanistic studies of various such reactions. However, so far natural isotope abundance determination of $O_2$ produced at solid electrode surfaces has not been applied. Here, we demonstrate that such measurements are possible. Moreover, they are experimentally simple and sufficiently accurate to observe significant effects. Our measured kinetic isotope effects depend strongly on the electrode material and on the applied electrode potential. They suggest that in the case of iron oxide as the electrode material, the oxygen evolution reaction occurs via a rate-determining O−O bond formation via nucleophilic water attack on a ferryl unit.

[1] Department of Chemistry and Pharmacy, Chemistry of Thin Film Materials, Friedrich-Alexander-Universität Erlangen-Nürnberg, Cauerstr. 4, 91058 Erlangen, Germany. [2] Department für Geographie und Geowissenschaften, GeoZentrum NordBayern, Applied Geology, Friedrich-Alexander-Universität Erlangen-Nürnberg, Schlossgarten 5, 91054 Erlangen, Germany. [3] Department of Chemistry and Institute of Materials Science, University of Connecticut, 55 North Eagleville Rd., Storrs, CT 06269, USA. [4] Institute of Chemistry, Saint Petersburg State University, Universitetskii pr. 26, Saint Petersburg, Russian Federation 198504. Correspondence and requests for materials should be addressed to A.M.A.-B. (email: alfredo.angeles-boza@uconn.edu) or to J.A.C.B. (email: johannes.barth@fau.de) or to J.B. (email: julien.bachmann@fau.de)

The direct electrochemical splitting of water into its constituent elements represents a non-polluting method of producing storable energy carrier from renewable sources, such as wind, hydroelectric or solar power. This power-to-fuel conversion involves two half-reactions: the four-electron, four-proton oxidation of water to dioxygen and the reduction of protons to dihydrogen[1,2]. The mechanistically more demanding and kinetically limiting half-reaction here is the generation of $O_2$, since it involves a proton and electron currency of four as well as the formation of the oxygen–oxygen double bond[3,4]. Thus, the overall water splitting activation energy requirement is defined, beyond the thermodynamic potential difference between the redox pairs ($\Delta E = +1.23$ V), mostly by the overpotential of the oxygen evolution reaction (OER, whereas the contribution associated to the overpotential of the hydrogen evolution reaction is significantly smaller)[2]. The list of materials evaluated for their potential use as OER catalysts is promising for the future and includes rare noble metal oxides[5], earth-abundant first-row transition metal oxides and (oxy)hydroxides[6–8]. However, de novo designs of completely original future catalysts must rely on the fundamental comprehension of possible OER mechanisms, including the unambiguous identification of their key (rate-determining) step. Thus, the elucidation of individual steps, intermediates and transition states represents one of the central challenges towards a broadly applicable power-to-fuel scheme via water electrolysis.

So far, studies on the identification of reaction intermediates at solid anode and photo-anode surfaces have relied on a variety of indirect techniques, such as electrochemical impedance spectroscopy[9], Tafel plot analysis[1,10], photoinduced absorption spectroscopy[11], time-resolved Fourier-transform infrared spectroscopy[12], attenuated total reflection infrared spectroscopy[13], Raman spectroscopy[14], and transient absorption spectroscopy[15,16], complemented by density functional theory (DFT) calculations[17,18].

Recently, iron oxide ($Fe_2O_3$) electrodes have received much attention by Peter[19,20], Hamann[9,13], Durrant[11,15,16], and their co-workers. In general, they have agreed on the first step of water oxidation involving the formation of an $Fe^{IV}=O$ species (ferryl) via oxidation of the initially hydroxylated $Fe^{III}$ surface (see mechanistic overview in Fig. 1). This is in part based on the similarity to other molecular systems amenable to investigation in a homogeneous (dissolved) phase. The steps following the ferryl formation are, however, still under discussion and probably vary depending on several parameters such as temperature, pH of electrolyte solution and nature of the electrolyte[11,13,18,21].

A direct approach to the identification of rate-determining steps (RDS) in molecular catalysis (such as the photosystem II complex, its models, or artificial OER catalysts) has been the analysis of kinetic isotope effects (KIEs)[22–29]: The mass of an atom affects the overall reaction kinetics if a bond in its vicinity is made or broken in the RDS. KIEs are largest, and have been exploited most extensively, in the case of $^2H/^1H$[30–32]. Isotope effects also exist for $^{18}O/^{16}O$. They directly probe the formation of the O−O bond and are therefore particularly well suited to the study of OER. However, they have been determined in a significantly smaller number of cases[25–27,33]. Determinations of $^{18}O$ KIE have shown that contrasting reaction steps are rate-determining in the photosynthetic activity of various microorganisms, which feature near-unity values of $^{18}O$ KIE (0.993 to 1.001) and in documented synthetic OER catalysts[22,24,34–36]. KIE determinations have been pivotal in refuting the original assumption that the O−O bond formation via symmetric coupling of two vicinal high-valent metal oxos is the RDS in catalysis by the blue dimer family of molecular ruthenium complexes[28,29,37–40].

Thus, the $^{18}O$ KIE could crucially contribute to the design of more suitable OER catalysts. However, the spectrum of its

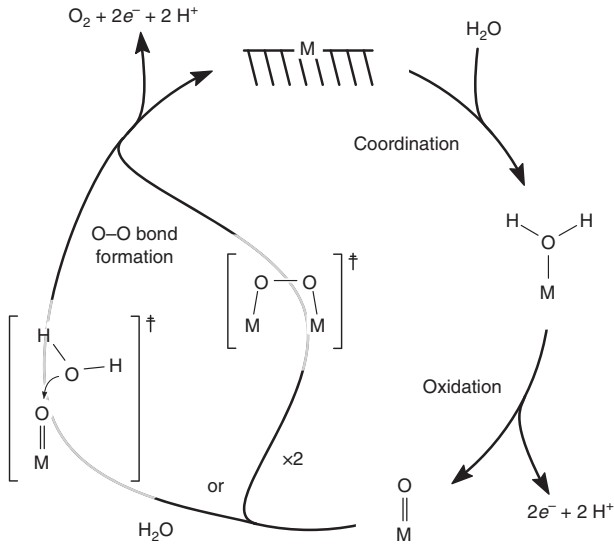

**Fig. 1** Minimal mechanism of the OER at a solid metal oxide electrode. While coordination of water is fast, the oxidation to a high-valent metal-oxo (which occurs in several steps) or the formation of an O−O bond (which can take place on either of two distinct pathways) can, depending on the system and conditions considered, represent the slowest and rate-determining step

practical applications has been limited by three significant experimental constraints. First, the number of $^{18}O$ KIE studies on heterogeneous systems is limited to a handful of examples with the most recent dating back to 1987[41–45]. Second, $^{18}O$ KIEs have mostly been determined via density measurements[41,42], later via mass spectrometric analysis of $CO_2$ after combustion[43] and rarely on the $O_2$ molecule itself. Third, isotopic labelling as usually performed for the determination of $^2H$ KIEs, is significantly less feasible as commercial sources are expensive[46].

The direct determination of small isotope composition changes on a natural-abundance level between a substrate and its product has been performed only in a very small number of cases in the field of photosynthetic water oxidation[22,34,47]. Two studies reported natural-abundance measurements by isotope ratio mass spectrometry (IRMS) directly of the $O_2$ product in heterogeneous (electrocatalytic) systems[44,45]. However, their experimental limitations have prevented the investigation of reaction mechanisms and/or RDS at solid OER electrocatalyst surfaces. (In one case, $BrF_3$ as the oxidant is described as the source of an unknown systematic error[44], whereas in the other study the KIE shows error bars of ±0.007 for variations from unity of ±0.025.)[45]

Here, we present a direct IRMS $^{18}O/^{16}O$ effect study of the OER at nanoporous $Fe_2O_3$ and Ir electrode surfaces under steady-state conditions in an aqueous electrolyte. These samples are chosen for their relatively high electrocatalytic activity (caused by the geometric surface area increase), which is necessary for the isotope analyses. We work at natural-abundance levels in continuous flow mode at $m/z = 32$ from small sample volumes (12 mL)[48], without preliminary conversion to $CO_2$[49,50]. The instrumental error on repeat standard measurements is ±0.0002. Repeat measurements of real samples yield a maximum standard deviation of ±0.0015. Experimentally determined changes of natural-abundance $^{18}O/^{16}O$ ratios of dissolved $O_2$ (DO) formed from water yield KIEs significant enough to differentiate between various catalysts and draw mechanistic conclusions. The data hint

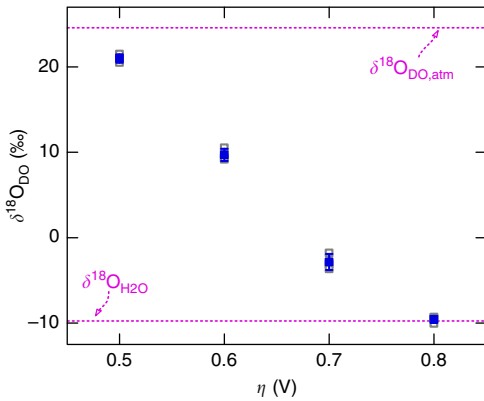

**Fig. 2** Potential-dependent $\delta^{18}O_{DO}$ composition of $O_2$ evolved at nanoporous $Fe_2O_3$ electrodes. The DO compositions are studied for $O_2$ evolving upon 3 h of steady-state electrolysis in a pH 7 aqueous $KH_2PO_4$ electrolyte at different applied overpotentials $0.48 \leq \eta \leq 0.78$ V (potentials $E$ vs. Ag/AgCl: $+1.10 \leq E \leq +1.40$ V). $\delta^{18}O_{DO}$ values are presented for three nominally identical individual samples (grey empty squares) and the corresponding average values with standard deviations (blue squares and bars). Isotopic compositions measured for atmospheric $O_2$ dissolved in the aqueous electrolyte and electrolyte $H_2O$ are given as control values (dashed lines): $\delta^{18}O_{DO,atm} = +24.6‰$, $\delta^{18}O_{H2O} = -9.8‰$

at a nucleophilic attack of water on a high-valent surface ferryl as the RDS of water oxidation at iron(III) oxide surfaces.

## Results

**Electrodes and electrolysis**. Nanoporous iron oxide and oxide-covered iridium electrodes were prepared by coating a highly ordered array of anodic alumina pores with a thin layer of approximately 10 nm $Fe_2O_3$ or 20 nm Ir via atomic layer deposition (ALD)[51–53]. These electrodes were used in steady-state electrolyses at different applied potentials, that is, different driving forces in pH 7 phosphate buffer (for $Fe_2O_3$) or acidic sulphuric acid electrolyte (for Ir). Let us focus on the $Fe_2O_3$ system first. Care was taken to exclude any contamination by atmospheric $O_2$. The electrochemical cell was degassed prior to the experiment and properly sealed as confirmed by a control experiment in which the working electrode was left in open circuit (Supplementary Fig. 1 in the Supplementary Information, dashed line). Upon electrolysis $O_2$ evolved continuously, quantified by an optical oxygen sensor. Supplementary Fig. 1 shows how the evolved $O_2$ concentration follows the Faradaic current for an applied potential $E$ of $+1.30$ V vs. Ag/AgCl (overpotential $\eta = 0.68$ V).

For subsequent IRMS analyses of the DO, 12 mL of electrolyte were extracted from the electrochemical cell and filled into helium-flushed vials with butyl rubber caps. Depending on the overpotential ($0.48 \leq \eta \leq 0.78$ V), the concentrations of DO evolved upon electrolysis obtained by IRMS lie between 5 and 50 µmol $L^{-1}$. They agree, within experimental uncertainty, with the values determined by an optical sensor (Supplementary Fig. 2). The measured $c_{DO}$ curve reveals the typical exponential dependence of electrocatalytic current density $J$ on applied overpotential $\eta$ (Supplementary Fig. 3). The values lie in the expected range based on the current passed and the partitioning of $O_2$ into the gas phase (see Supplementary Note 1).

**Isotope composition of evolved oxygen**. The isotope composition of DO in the aqueous electrolyte is expressed by ratios of the stable oxygen isotopes $^{18}O$ and $^{16}O$ in the standard $\delta$-notation in per mille (‰) normalised to Vienna Standard Mean Ocean Water

(VSMOW)[36]. As controls, IRMS analyses of aerobic $O_2$ dissolved in the pH 7 phosphate electrolyte and of the associated $H_2O$ yielded $\delta^{18}O_{DO,atm}$ and $\delta^{18}O_{H2O}$ values of $+24.6$ and $-9.8‰$, respectively ($+24.7$ and $-9.6‰$ in the sulphuric acid electrolyte). These values agree with the literature[36,54–56]. Figure 2 shows the influence of applied overpotentials on the isotopic compositions of the evolved DO after 3 h of electrolysis. From a starting value of $\delta^{18}O_{DO} = 21.0‰$ at 0.48 V, a significant decrease in $\delta^{18}O_{DO}$ is observed down to $-9.6‰$ at $\eta = 0.78$ V. Thus, the applied overpotential (i.e., the external activation energy) has a profound influence on $\delta^{18}O_{DO}$.

In control experiments, we checked that the electrolysis duration has negligible influence on the isotopic composition of the product (Supplementary Fig. 4), proving again the lack of adventitious atmospheric $O_2$. Furthermore, the fractionation effect worth 0.7‰ of physical $O_2$ transfer from the dissolved to gaseous phase[57] is not significant with respect to the effects measured experimentally. Thirdly, taking into account the $O_2$ contamination visible in Supplementary Fig. 1 (DO concentration of 3 µmol $L^{-1}$ or 0.1 ppm after degassing) would affect our $\delta^{18}O_{DO}$ value by a maximum of $-1.9‰$ (the detailed calculation and values are presented in the Electronic Supplementary Information, included in Supplementary Table 1 and Supplementary Note 2). This number is significantly smaller than the values of $\delta^{18}O_{DO}$ and differences between them determined in this work. Thus, measuring the DO is representative of the whole $O_2$ generated at the electrode. The fourth and final possible source of error that we considered is the possibility that $O_2$ produced at the working electrode be reduced back to water at the counter-electrode. This would affect the isotopic composition of DO in an uncontrolled manner. This possibility is excluded by our experimental measurement of the potential difference between working electrode and counter-electrode: since the potential difference is larger than $+1.10$ V in all conditions of bulk electrolysis used here, we can positively conclude that the reaction occurring at the counter-electrode is the reduction of water to $H_2$.

The experiment performed on iridium electrodes shows a qualitatively similar trend of $\delta^{18}O_{DO}$ decline with $\eta$ increase as observed for the $Fe_2O_3$ electrode (Supplementary Fig. 5). At low overpotentials, the $\delta^{18}O_{DO}$ values are more positive than the substrate (water). Thus, both catalysts preferentially consume $H_2^{18}O$. At large applied $\eta$, both catalysts converge to a $\delta^{18}O_{DO}$ almost identical to that of the substrate $H_2O$ consumed in the reaction. In other words, we observe that the discrimination of the catalyst for $H_2^{18}O$ is reduced with increasing overpotential. This is typical of electrochemical reactions, where the applied overpotential not only affects the thermodynamic driving force of the reaction, but also its activation energy. In fact, this effect furnishes the basis of the Butler–Volmer equation, the fundament of all electrochemical kinetics. When two reactions have distinct activation energies at equilibrium ($\eta = 0$, Fig. 3 left), applying an overpotential ($\eta > 0$) reduces both of their activation energies, which simultaneously and necessarily also reduces the difference between their activation energies (Fig. 3 centre and right). In other words, the selectivity of electrochemical reactions decreases with increasing overpotential without a change in the reaction mechanism (a quantitative treatment of this effect in the framework of Marcus' theory of electron transfer has been published)[58]. Thus, for our case, the properties of electrochemical reactions cause the discrimination between $^{18}O$ and $^{16}O$ to decrease as $\eta$ is increased: Accordingly, the approach towards the $\delta^{18}O_{H2O}$-value observed at high overpotentials is plausible, because after a critical point the driving force has become so large that any activation energy is insignificant. In these conditions of large overpotentials, the electrochemical turnover at the surface is large and not selective for stable isotopes and

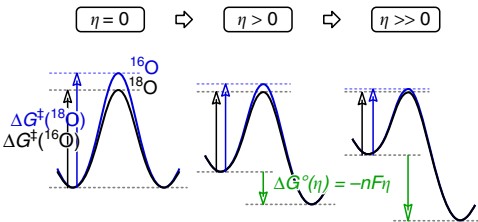

**Fig. 3** Schematic explanation for the decrease of selectivity upon increase in overpotential in electrochemical reactions. A change in mechanism is not required to explain a decrease in selectivity. The scheme includes the free enthalpy of activation $\Delta G^{\ddagger}$, and the free enthalpy of reaction $\Delta G^{\circ}$, which depends on the number of electrons $n$ exchanged in the rate-determining step, the Faraday constant $F$ and the overpotential $\eta$

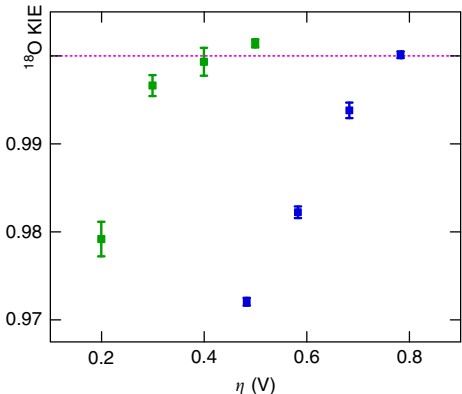

**Fig. 4** Potential-dependent $^{18}O$ KIEs for the OER at nanoporous $Fe_2O_3$ or Ir electrodes. $^{18}O$ KIEs of nanoporous $Fe_2O_3$ (blue data points) or Ir (green data points) are presented for different applied overpotentials $\eta$. Steady-state electrolyses were performed in a 0.1 M $KH_2PO_4$ solution (pH neutral) for 3 h or in a 0.1 M $H_2SO_4$ solution for 0.5 h at $Fe_2O_3$ and Ir electrodes, respectively. The $^{18}O$ KIEs presented are average values calculated from three nominally identical individual samples. Their corresponding standard deviations are shown as error bars. The unity KIE value is indicated by the dashed line

diffusion becomes rate-limiting. Thus, the isotopic composition of $H_2O$ is directly transferred to its product $O_2$.

**Mechanistic interpretation**. For further interpretation, we convert the abundance ratios $R_{O2}$ of $O_2$ produced from natural-abundance water to $^{18}O$ KIEs, specifically, the ratio between the rate constants $k$ for conversion of $H_2^{16}O$ and $H_2^{18}O$ [59,60]. The $^{18}O$ KIEs determined for $Fe_2O_3$ electrodes (blue data points) at overpotentials $0.48 \leq \eta \leq 0.78$ V (Fig. 4) range between 0.970 and 1.000. When translated from the isotope value above, the unity KIE value at large $\eta$ corresponds to the absence of isotopic discrimination when the reaction becomes diffusion-controlled. More importantly, the $^{18}O$ KIEs determined at $\eta \leq 0.68$ V are all KIE < 1.000 (the values lie between 0.970 and 0.993). This is usually referred to as an 'inverse' KIE[59]. Taking into account the fractionation effect of 0.7‰ mentioned above for the physical transfer of $O_2$ from the dissolved to gaseous phase causes a maximal variation of $^{18}O$ KIEs by 0.0007. The effect of adventitious DO (also mentioned above) represents 0.0018 in KIE units. It is possible that this could affect the absolute values somewhat, yet the trend remains unambiguous.

The completely different electrocatalyst iridium also yields a pronounced inverse KIE with $^{18}O$ KIEs ranging between 0.977 and 1.001 for overpotentials $0.20 \leq \eta \leq 0.50$ V (green data points). However, a significant difference is that the noble metal

electrodes approach $^{18}O$ KIE unity much faster than their 'rust' counterparts. This behaviour is likely due to the higher catalytic activity of Ir. For this catalyst the activation energy is lower to start with, the case of negligible discrimination is reached much earlier. The shift of the curves is about 0.4 V, or 39 kJ mol$^{-1}$. This difference can be interpreted as the additional activation energy decrease that could be achieved by the Ir surface with respect to $Fe_2O_3$.

For a mechanistic interpretation of the isotope effect, let us now focus on the $Fe_2O_3$ electrodes, for which ample literature material can be exploited[9,11,13,16,19]. Additional information is provided by $^2H$ isotope effects:[46] The use of deuterated water significantly decreases the OER activity (Supplementary Fig. 6 and Supplementary Table 2). The ratio of exchange current densities $J_0$ for OER in $H_2O$ and $^2H_2O$ is determined from the quasi-steady-state current-potential curve (Tafel slopes, see Methods section and Supplementary Fig. 6). This ratio is equal to the ratio of rate constants $k_0$ and defines the $^2H$ KIE. Using the Tafel slope over the range $0.45$ V $\leq \eta \leq 0.55$ V, the calculated $^2H$ KIE is 0.56 (varying the $\eta$-range for the fit yields KIE values between 0.56 and 0.96, always inverse).

## Discussion

What is the mechanistic origin of the inverse $^{18}O$ KIE determined during turnover conditions for OER? This question can be answered by the competitive $^{18}O$ KIEs, which yield information about the steps from the encounter of the substrate ($H_2O$) with the catalyst up to the first irreversible step[61]. Regardless of the specific mechanism in OER for the catalysts under study, more than one step is expected to be isotopically sensitive. Interestingly, $^{18}O$ KIEs have been shown to report on O−O bond formation steps as well as rate-limiting steps that occur before O−O bond formation in water oxidation reactions catalysed by metal ions[25,27,62]. Both the O−O bond formation and its preliminary steps have been hypothesised as rate-limiting steps for various systems[1,11,13,17,21,63].

In order to test these two hypotheses in our iron oxide system, we analyse the isotope effect data considering the simplest possible mechanism for $O_2$ evolution during the catalytic cycle (Fig. 5). Binding of water molecules is known to be a reversible process and therefore we do not consider it as a potential rate-limiting step in our analysis[44]. The following discussion will be based on the closest possible systems for which isotope effect values are available in the literature.

First, we consider the hypothesis that the oxidation/deprotonation is the primary irreversible step. In this case, the overall $^{18}O$ KIE determined ($KIE_{obs}$) represents the product of the equilibrium isotope effect (EIE) for $H_2O$-binding ($EIE_{coord}$) and the intrinsic KIE for the oxidation/deprotonation step ($KIE_{ox}$) shown in Eq. 1:

$$KIE_{obs}^{ferryl} = EIE_{coord} \cdot KIE_{ox} \qquad (1)$$

Given the preference of metal ions for $H_2^{18}O$ over $H_2^{16}O$, the binding of water to the surface is expected to produce an inverse $EIE_{coord}$[64,65]. This effect has been estimated by Taube to be in the range of 0.95−0.97[44]. Given our experimentally determined $KIE_{obs}$ of 0.97 at low overpotential, the expected $^{18}O$ KIE for the oxidation/deprotonation process must be in the range of $1.00 \leq KIE_{ox} \leq 1.02$. This is inconsistent with $KIE_{ox}$-values determined for the oxidation of aquo Fe(II) ions, which vary from $1.04 \pm 0.01$ to $1.09 \pm 0.02$ as supported by a nonadiabatic multiphonon quantum mechanical approach[66]. Furthermore, rate-determining oxidation/deprotonation results in small normal $^2H$ KIE values (1.03−1.07)[25–27], which contradicts the observations for the

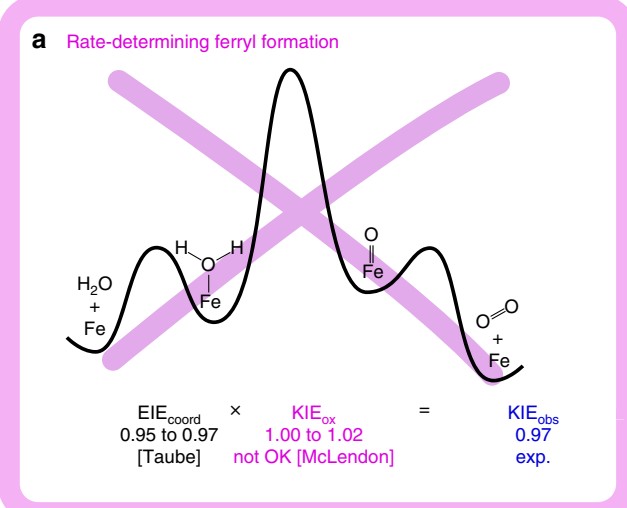

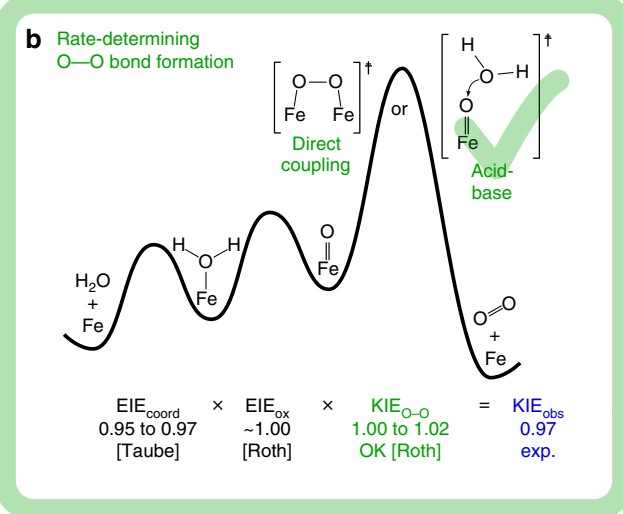

**Fig. 5** Qualitative energy diagrams for the mechanistic interpretation of the observed $^{18}O$ KIE. The mechanistic interpretation is provided in the framework of the minimal reaction mechanism sketched in Fig. 1. It shows two distinct reaction steps as potentially rate-determining:[1,11,13,17,21,63] either **a** the local oxidation and deprotonation to a reactive metal-oxo or **b** the O−O bond formation with two distinct possible mechanisms. For the first scenario, the $^{18}O$ KIE determined experimentally (KIE$_{obs}$) is not consistent with the range of values expected from literature data, whereas for the latter case KIE$_{obs}$ is in agreement with the data available so far, indicating an acid-base mechanism rather than direct oxo coupling[25,27,44,66]. The overall $^{18}O$ KIE determined (KIE$_{obs}$) represents the product of the equilibrium isotope effect (EIE) for $H_2O$-binding (EIE$_{coord}$) and the intrinsic KIE for the oxidation/deprotonation step (KIE$_{ox}$)

$Fe_2O_3$ system being studied. Thus, our experimental data allow us to exclude conclusively that oxidation/deprotonation is the RDS.

In the alternative scenario with the O−O bond formation as the rate-limiting step, the observed KIE results from Eq. 2,

$$\text{KIE}_{obs}^{O-O} = \text{EIE}_{coord} \cdot \text{EIE}_{ox} \cdot \text{KIE}_{O-O} \qquad (2)$$

where EIE$_{ox}$ represents the EIE for the oxidation/deprotonation step and KIE$_{o-o}$ is the KIE of the O−O bond formation step. To our knowledge, EIE$_{ox}$ has not been directly determined for iron-containing systems, but it can be estimated to be ~1.00 based on computational calculations of three different ruthenium systems that catalyse the oxidation of water[27]. Thus, given $0.95 \leq \text{EIE}_{coord} \leq 0.97$ and $\text{KIE}_{obs} = 0.97$, the KIE$_{o-o}$ is in the $1.00-1.02$ range, equating to a normal KIE. Normal $^{18}O$ KIEs have been determined to originate from O−O bond formation steps in four different homogeneous catalysts, including ferrate[25], which agrees with this estimation. Further credence to this model is provided by the inverse $^2H$ KIE values $(0.10-0.95)$ determined for these systems[25], which coincide with our $^2H$ KIE values of $0.56-0.96$. Additionally, experimental evidence by Hamann et al. identified a surface ferryl as an intermediate species in line with our interpretation[13].

Van Voorhis et al. have recently predicted the O−O bond formation as the rate-limiting step for iron oxide, based on DFT calculations in combination with Sabatier analysis[17]. In these calculations, both the acid-base and direct coupling of metal-oxo species seem to be equally plausible. In the four systems in which $^{18}O$ KIEs were used to study O−O bond formation experimentally, it has been reported that direct coupling of metal-oxo species leads to $^{18}O$ KIE values larger than $1.030$[27]. In contrast, acid-base mechanisms range from 1.017 to 1.031[27]. Based on the range of possible $^{18}O$ KIE$_{o-o}$ for the system studied here $(1.00-1.02)$ we suggest that the rate-limiting step in OER catalysed by $Fe_2O_3$ is the O−O bond formation via an acid-base mechanism rather than a direct coupling mechanism.

In conclusion, direct natural-abundance determinations by IRMS of product isotope compositions $(\delta^{18}O_{DO})$ are experimentally facile to implement. Macroscopic $\delta^{18}O_{DO}$ values can easily be converted to KIEs. The overall kinetic isotope effects (KIE$_{obs}$) determined in this manner are sufficiently accurate to allow for microscopic, mechanistic interpretation. The application of this method to heterogeneous electrocatalytic water oxidation systems yields clear dependencies on the applied overpotential and distinct profiles for two different electrode materials. The KIE$_{obs}$ in the iron oxide case is quantitatively incompatible with the high-valent ferryl formation as the RDS. Instead, it favours the hypothesis of a nucleophilic water attack on the ferryl as the mechanism of the RDS over the direct coupling between two adjacent ferryl units. The methods demonstrated in this paper render systematic investigations of various heterogeneous or heterogenised electrocatalysts (and photoelectrocatalysts) possible. We envision such systematic studies will lead to the identification of distinct catalyst classes governed by distinct reaction mechanisms. Such information has not been available so far. The insight generated will lay the foundation for a more efficient search of improved or alternative water oxidation catalysts.

## Methods

**Materials**. Chemicals were purchased from Sigma-Aldrich, Alfa Aesar, ABCR, or VWR and used as received. Water was purified in a Millipore Direct-Q system for application in electrolytes. Aluminium plates (99.99%) and Si (100) wafers covered with an oxide layer were supplied by Smart Membranes and Silicon Materials Inc., respectively. Ozone was generated with a BMT 803N ozone generator from oxygen purchased from Air Liquide.

**Preparation of iron oxide electrodes**. Nanoporous iron oxide electrodes were prepared according to previously published work on porous $Fe_2O_3$ electrodes[52]. Anodic aluminium oxide (AAO) of approximately 11 μm length (obtained with 4 h of anodization duration during the second anodization step) and 370 nm diameter were used as porous templates. One side of the nanostructured substrate was closed with a several micrometre-thick nickel layer, serving as the electrical backside contact in electrochemical measurements. The porous templates were then coated with $Fe_2O_3$ via ALD in a commercial Gemstar-6 XT ALD reactor from Arradiance operating with a $N_2$ carrier gas. The deposition was performed at 200 °C with ferrocene (FeCp$_2$, kept in a stainless steel bottle maintained at 45 °C) and ozone as reactants. In the case of ferrocene, a 2-s pulse with 40 s of subsequent exposure time was repeated two times before the chamber was purged with $N_2$ for 90 s. For

$O_3$, single pulses were carried out with pulse, exposure and purge times of 0.5, 40, 90 s, respectively. Eight hundred ALD cycles yielded a thickness of approximately 12 nm of $Fe_2O_3$. A spectroscopic ellipsometer SENpro by Sentech was used to determine the $Fe_2O_3$ thickness on an $Al_2O_3$-coated silicon wafer, which was added to the reaction chamber. A total of 50 data points were recorded for wavelengths between 380 and 1050 nm under an angle of incidence of 70° with a tungsten halogen lamp as light source. The data orientation $\Theta$ and ellipticity $\varepsilon$ were then fitted with a fixed optical model for ALD-$Fe_2O_3$ with SpectraRay/3.

**Preparation of iridium electrodes**. Nanoporous iridium electrodes were produced by the multistep preparation procedure by Schlicht et al.[53]. The pore length and diameter of the AAO template used here were approximately 17 μm (obtained with 8 h of anodization duration) and 370 nm, respectively. The porous substrates, also equipped with an electrical nickel backside contact, were coated with a thin iridium layer in the same commercial Gemstar-6 XT ALD reactor. The deposition process was carried out at 220 °C with ethylcyclopentadienyl-1,3-cyclohexadiene-iridium (I) ((EtCp)Ir(CHD) from Strem, kept at 90 °C in a stainless steel bottle) and ozone as precursors. Four consecutive microcycles consisting of a 0.5 s (EtCp)Ir(CHD) pulse and 40 s of exposure time were performed before the chamber was purged with $N_2$ for 90 s. For $O_3$, single pulses were carried out with pulse, exposure and purge times of 0.5, 40, 90 s, respectively. Approximately 20 nm of iridium were deposited with 240 ALD cycles. An indium-doped tin oxide-coated silicon wafer was used as a reference for the layer thickness determination via spectroscopic ellipsometry.

**Electrochemical studies**. The nanoporous electrodes were laser-cut into half with a GCC LaserPro Spirit LS Laser and glued with the nickel contact on smaller copper plates ($A \approx 4$ cm$^2$) with double-sided conductive copper foil (Supplementary Fig. 7a). A chemically resistant and electrically insulating polyimide (Kapton) adhesive tape, featuring a laser-cut oval ($a = 0.3$ cm and $b = 0.5$ cm), was used to define the macroscopic sample area $A = 0.471$ cm$^2$ exposed to the electrolyte also protecting the copper plate from the electrolyte. All electrochemical studies were performed at room temperature in a custom-made glovebox continuously flushed with $N_2$ using a Gamry Interface 1000 potentiostat. Supplementary Figure 7b shows nanoporous working electrodes (WE) adjusted into a double-walled cell (EuroCell from Gamry instruments) that was additionally equipped with a graphite rod as the counter-electrode (CE), a Ag/AgCl/KCl(sat.) reference electrode (RE, shifted by +0.20 V relative to the normal hydrogen electrode), and an optode (model Visiferm DO 160 by Hamilton) for direct DO quantification with an accuracy ±0.05 vol.%. The cell was filled with 130 mL of fresh electrolyte, properly sealed and degassed with $N_2$ for 1 h via a septum prior to each measurement. A 0.1 M aqueous $KH_2PO_4$ solution (adjusted to pH 7) and a 0.1 M $H_2SO_4$ solution (pH 0.5) were used as electrolytes for nanoporous $Fe_2O_3$ or iridium electrodes, respectively.

**Electrochemical studies of the $^2H$ kinetic isotope effect**. The effect of protons on the OER was investigated at nanoporous $Fe_2O_3$ electrodes. Cyclic voltammograms were therefore measured at a scan rate of 5 mV s$^{-1}$ in protio and deuterio ($^2H_2O$, 99.9 at.%) solutions containing 0.1 M $KH_2PO_4$. Both phosphate electrolytes were adjusted to pH 7 with 5 M NaOH dissolved in $H_2O$ or $^2H_2O$, respectively.

**Preparation of water samples for KIE analysis**. The oxygen isotope fractionation and the corresponding $^{18}O$ KIE of the water oxidation reaction was investigated for $Fe_2O_3$ and iridium as catalyst materials. Steady-state electrolyses were performed under stirring for various reaction conditions. In the case of water oxidation at nanopore $Fe_2O_3$ electrodes, either the applied potential or the duration was varied between $+1.10 \leq E \leq +1.40$ V (overpotentials $0.48 \leq \eta \leq 0.78$ V) and $0.50 \leq t \leq 5.25$ h, respectively. For iridium electrodes, steady-state electrolyses were conducted for a fixed duration ($t = 0.5$ h) in a potential range of $+1.20 \leq E \leq +1.50$ V (overpotentials $0.20 \leq \eta \leq 0.50$ V). After termination of each experiment, aqueous electrolyte was directly extracted with a syringe from the electrochemical cell and transferred into helium-flushed vials ($V = 12.1$ mL). For proper sealing, each vial was completely filled with electrolyte solution and immediately capped with a butyl rubber septum held in place with a plastic screw cap. These aqueous samples were then analysed within 24 h for their $\delta^{18}O_{DO}$. One concern when conducting this experiment was leakage of atmospheric $O_2$ into the sample vials that would have masked the oxygen produced by the OER. Any contributions of atmospheric oxygen due to improper sealing could be excluded via a series of blank samples that revealed negligible incoming $O_2$. In addition, samples for stable oxygen isotope determinations of $H_2O$ (expressed as $\delta^{18}O_{H2O}$) in the non-converted (fresh) electrolyte were also studied as reference values.

**Mass spectrometric analysis of oxygen isotopes**. All values are reported in the standard delta notation in per mille (‰) versus the VSMOW according to Eq. 3

$$\delta^{18}O = \left[ \frac{R_{sample}}{R_{standard}} - 1 \right] \times 1000 \text{ ‰} \tag{3}$$

where $R$ is the molar ratio of the heavy and light isotope of an element ($n(^{18}O)/n$

($^{16}O$)) in the sample and the standard (VSMOW), with the latter having a value of $2.0052 \times 10^{-3}$[67,68]. Stable oxygen isotopes of DO (expressed as $\delta^{18}O_{DO}$) were measured with an automated continuous flow approach by Barth et al[48]. that was adapted to a Gasbench II auto sampler coupled to a Thermo Fischer Delta V Advantage IRMS. The isolation of DO into a headspace relies on a helium extraction technique by Wassenaar and Koehler[69]. In the adapted technique, a headspace of $3-6$ mL was generated by removing the same aliquot of water via a double syringe system. Variable headspaces were necessary to adapt to various DO concentrations. The DO was then transferred to the headspace by shaking the vials for 30 min on an orbital shaker. Then the headspace was extracted into a sampling loop on the Gasbench II system. From there it was passed on to the IRMS in a helium flow via a capillary gas chromatographic column that separated $O_2$ from $N_2$ and accessory gases (e.g. Ar). The GC column was a Varian CP molecular sieve column of 25 m length with an inner diameter of 0.53 mm. It had an inner film of 0.5 μm thickness that acted as a 5 Å molecular sieve. The entry pressure of the helium carrier gas was 0.6 bar, yielding a flow of 5.6 mL min$^{-1}$. All operations were performed at 25 °C. This process yields sufficient $O_2$ gas from 12-mL water samples for reliable IRMS analyses. The additional tuning of the mass spectrometer to an $m/z$ ratio of 32 enables favourable direct determination of molecular $O_2(g)$ without combustion to $CO_2$[49,50]. The standard deviation of this method was determined with a value better than ±0.2‰ vs. VSMOW by multiple measurements of $\delta^{18}O_{DO}$ controls and standards.

The $\delta^{18}O_{H2O}$ compositions were analysed by an automated equilibration unit (Gasbench II) that was coupled in continuous flow mode to a Delta plus XP IRMS by Thermo Fisher Scientific. External reproducibility based on repeat analyses of laboratory internal controls was better than 0.1‰. All data for stable isotope measurements were corrected for a blank signal, linearity (i.e. detector-related shifts of isotope ratios in response to peak size) and instrumental drift during the run[70].

All $\delta^{18}O_{DO}$ values were further converted into abundance ratios $R$ in order to define $^{18}O$ KIEs[59,60]. If the substrate $H_2O$ can be treated as an infinitely large reservoir with insignificantly small molar changes upon product ($O_2$) formation[60], then Eq. 4 is valid:

$$\text{KIE}\left(^{18}O\right) = \frac{k_{light}}{k_{heavy}} = \frac{k_{16O}}{k_{18O}} = \frac{k_{substrate}}{k_{product}} = \frac{k_{H_2O}}{k_{O_2}} \tag{4}$$

with $R_{H2O} = 1.986 \times 10^{-3}$ for the aqueous electrolyte.

## Data availability
The data that support the findings of this study are available from the corresponding author upon reasonable request.

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

## Acknowledgements

We thank the Friedrich-Alexander-Universität Erlangen-Nürnberg for travel support in the Visiting Professor programme. A.M.A.-B. is grateful for support from the National Science Foundation CAREER grant (CHE-1652606). The work was also supported by the DFG grant IsoDO (BA 2207/15-1).

## Author contributions

The data were interpreted and the manuscript written with contributions of all authors. S.H. and S.S. prepared the samples and performed the $O_2$ sampling upon electrolysis. M. M. analysed the dissolved oxygen by IRMS and determined the isotope compositions. A. M.R. contributed to discussions and editing. A.M.A.-B., J.A.C.B., and J.B. devised the project.

## Additional information

**Competing interests:** The authors declare no competing interests.

