## [Peer Review File · Nature Communications]

Reviewers' comments:

Reviewer #1 (Remarks to the Author):

The manuscript by Haschke et al. reports ^{18}O kinetic isotope effects of water oxidation catalyzed by an electrode coated with iron oxide or iridium oxide. Both systems show inverse KIEs around 0.97 at low overpotential, and KIEs = 1.00 at high overpotential. The results are assigned to a rate-determining step of O-O bond formation via nucleophilic water attack on a ferryl unit, rather than ferryl formation. The KIE measurements appear to be well done, but the interpretation of the overpotential dependence of the KIE data is not compatible with the proposed mechanism. Therefore, the manuscript is not acceptable for publication in its current form.

1. [Line 150] The authors state that the driving force becomes so large at high overpotentials that the activation energy is insignificant. However, higher overpotentials merely provide higher $\text{Fe}=\text{O}$ concentrations at the electrode, according to the proposed mechanism. The reaction would go faster, but the KIE, intrinsic to the water nucleophilic attack chemistry, shouldn't be affected. There must be a change of mechanism with increasing overpotential, or an overpotential-dependent fractionation that leads to this result. The authors should address the discrepancy between this interpretation and the proposed mechanism, before this paper can be acceptable.

2. In general, the mechanistic analysis is not very convincing. The authors use data from other studies to make mechanistic arguments, but these systems are not closely similar. Moreover, it is not known what structure on the heterogeneous metal oxide surface is involved in the catalytic reaction. Hence, the statement in the abstract that the results "provide solid evidence that ... the oxygen evolution reaction occurs via a rate-determining O-O bond formation" is not supported. In fact, the statement on line 239 is more qualified: "we suggest that the rate-limiting step in OER ... is the O-O bond formation".

^{18}O KIE is a powerful approach to probe the RDS of a reaction, providing valuable mechanistic information. However, unlike the H/D KIE, the ^{18}O KIE is much smaller, and any physical or chemical event that involves oxygen may lead to a fractionation, changing the final KIE result. The authors do a good job excluding possible contamination by atmospheric oxygen by running the reaction in a glove box. However, the authors should consider the following points regarding possible fractionations of the oxygen isotope composition.

3. [Line 100] The authors take only dissolved oxygen into account. It is reported to be a 0.7‰ difference in $\delta^{18}\text{O}$ between dissolved and gaseous oxygen at equilibrium [Benson and Krause (1980)]. This is not a big effect, but it would be judicious to include it in the error estimation. This effect is not negligible compared to the reported error of 0.0015 KIE unit.

4. According to Figure S1, at the beginning of the electrolysis, the CDO is not zero ($\sim 3 \mu\text{mol/L}$). It actually looks like a significant amount (probably $>10\%$) compared to the final CDO at 4 hr ($\sim 27 \mu\text{mol/L}$). The DO contamination at 1 hr needs to be corrected to obtain accurate KIE results.

5. Oxygen reduction to water at the counter electrode is possible, and the oxygen consumption leads to a change to the isotope composition of the remaining oxygen. Do the authors assume that this effect is negligible?

In addition, the authors should consider the following two points regarding the supplementary information.

6. The black dashed line in Figure S1 should be changed into blue to be consistent with the left axis color.

7. It should be Figure S7, rather than Figure 7, in the text of pages S9-S10.

Reviewer #2 (Remarks to the Author):

This is an excellent paper describing the kinetic isotope effect of water oxidation with two representative metal oxide surfaces to glean mechanistic insight. The work focuses on iron oxide as it has been the subject of significant research and thus ample supporting literature but also includes comparison data on iridium oxide. The paper is well written and the work and analysis is thorough. I recommend publication and only have a couple of minor points for the authors to consider.

1. on page 1, line 37, the authors claim that only the thermodynamic potentials and oxygen evolution overpotential define the overall potential required for water splitting. what about the hydrogen evolution reaction? Even with the best catalyst (Pt) there is some overpotential which should be accounted for.

2. Is there a reason the authors chose to fabricate the electrodes the way they did, with alumina templates and ALD post modification? This seems to be an overly complicated synthesis of iron oxide electrodes. This is certainly a large barrier for others to try and reproduce the results. If the results (as presented) are general, and not specific to the electrode preparation, then why not use a simpler system? Some comment about this in the start of the results section would be helpful.

3. figure 4 and related text. IR measurements have indicated a Fe=O intermediate species of water oxidation with iron oxide (the authors reference 16). Observation of an intermediate species must come before the rate determining step to be observed. This supports the authors conclusion of rate-determining O-O bond formation and is worth noting.

Reviewer #3 (Remarks to the Author):

The publication „Direct oxygen isotope effect identifies the rate-determining step of electrocatalytic oxygen evolution reaction at a metal oxide surface“ by Haschke et al. describes a very interesting and new approach to investigate the oxidation of water at electrode materials. Apart from other in the article mentioned methods to investigate the involved processes, this new method provides a much deeper view on the involved processes. Especially, in the field of finding new electrode materials, this method will be useful. The publication is well structured and written in good English. Isotope data are well and clear presented and the statistical evaluation of the data is good. Due to the novelty and for discussion of the method in a broader scientific community, I recommend a publication in Nature Communications.

Detailed responses to the Reviewers' comments and description of actions taken

Reviewer #1

The manuscript by Haschke et al. reports ^{18}O kinetic isotope effects of water oxidation catalyzed by an electrode coated with iron oxide or iridium oxide. Both systems show inverse KIEs around 0.97 at low overpotential, and KIEs = 1.00 at high overpotential. The results are assigned to a rate-determining step of O-O bond formation via nucleophilic water attack on a ferryl unit, rather than ferryl formation. The KIE measurements appear to be well done, but the interpretation of the overpotential dependence of the KIE data is not compatible with the proposed mechanism. Therefore, the manuscript is not acceptable for publication in its current form.

We are convinced that the additional data and explanation provided in our revised version (see details below) can now convince the Reviewer that our interpretation is solid.

1. *[Line 150] The authors state that the driving force becomes so large at high overpotentials that the activation energy is insignificant. However, higher overpotentials merely provide higher Fe=O concentrations at the electrode, according to the proposed mechanism. The reaction would go faster, but the KIE, intrinsic to the water nucleophilic attack chemistry, shouldn't be affected. There must be a change of mechanism with increasing overpotential, or an overpotential-dependent fractionation that leads to this result. The authors should address the discrepancy between this interpretation and the proposed mechanism, before this paper can be acceptable.*

In electrochemistry, a change in mechanism is not required to explain a reduction in fractionation (or selectivity, as it would be called in catalysis). Indeed, the fact that activation energies are reduced by an increase in overpotential naturally leads to such a decrease in fractionation. We have added a detailed explanation of this phenomenon to our revised text, including a figure which renders this effect intuitive to grasp and a reference to a quantitative treatment of it.

2. *In general, the mechanistic analysis is not very convincing. The authors use data from other studies to make mechanistic arguments, but these systems are not closely similar. Moreover, it is not known what structure on the heterogeneous metal oxide surface is involved in the catalytic reaction. Hence, the statement in the abstract that the results "provide solid evidence that ... the oxygen evolution reaction occurs via a rate-determining O-O bond*

formation" is not supported. In fact, the statement on line 239 is more qualified: "we suggest that the rate-limiting step in OER ... is the O-O bond formation".

We have revised the wording of the statement in the abstract and replaced "provide solid evidence" with "suggest". We have also added a word of caution to our discussion.

18O KIE is a powerful approach to probe the RDS of a reaction, providing valuable mechanistic information. However, unlike the H/D KIE, the 18O KIE is much smaller, and any physical or chemical event that involves oxygen may lead to a fractionation, changing the final KIE result. The authors do a good job excluding possible contamination by atmospheric oxygen by running the reaction in a glove box. However, the authors should consider the following points regarding possible fractionations of the oxygen isotope composition.

3. *[Line 100] The authors take only dissolved oxygen into account. It is reported to be a 0.7‰ difference in d18O between dissolved and gaseous oxygen at equilibrium [Benson and Krause (1980)]. This is not a big effect, but it would be judicious to include it in the error estimation. This effect is not negligible compared to the reported error of 0.0015 KIE unit.*

We have calculated the potential error caused by this effect, which amounts to 0.0007 in KIE units. This information has been added into our text (lines 136 and 188). This potential systematic error is smaller than the experimentally determined random error of 0.0015, but indeed not insignificant with respect to it.

4. *According to Figure S1, at the beginning of the electrolysis, the CDO is not zero (~3 μmol/L). It actually looks like a significant amount (probably >10%) compared to the final CDO at 4 hr (~27 μmol/L). The DO contamination at 1 hr needs to be corrected to obtain accurate KIE results.*

Good point. We have also implemented this calculation to a new small paragraph dedicated to the various potential sources of error (lines 136 to 149). We have come to the conclusion that the maximal error potentially caused by contamination by these 3 μmol/L (or 0.1 ppm!) corresponds to -1.9‰ in δ¹⁸O, or 0.0018 in KIE units. This would indeed affect the absolute values somewhat, yet the trend remains unambiguous. Moreover, these -1.9 ‰ should have affected all results in the same manner. Therefore, this effect should cancel out.

5. *Oxygen reduction to water at the counter electrode is possible, and the oxygen consumption leads to a change to the isotope composition of the remaining oxygen. Do the authors assume that this effect is negligible?*

We have also performed additional experiments to address this question, and have added their results to the "error paragraph", as well. We find that the potential of

the counter- electrode is that of the $\text{H}_2\text{O}/\text{H}_2$ couple and not that of $\text{O}_2/\text{H}_2\text{O}$. Therefore, the reaction occurring at the counter-electrode is predominantly the reduction of water to H_2 .

In addition, the authors should consider the following two points regarding the supplementary information.

6. *The black dashed line in Figure S1 should be changed into blue to be consistent with the left axis color.*

Thank you for the attentive reading. This remark has been implemented.

7. *It should be Figure S7, rather than Figure 7, in the text of pages S9-S10.*

Thank you again. This correction has been performed.

Reviewer #2

This is an excellent paper describing the kinetic isotope effect of water oxidation with two representative metal oxide surfaces to glean mechanistic insight. The work focuses on iron oxide as it has been the subject of significant research and thus ample supporting literature but also includes comparison data on iridium oxide. The paper is well written and the work and analysis is thorough. I recommend publication and only have a couple of minor points for the authors to consider.

Thank you. We have implemented all of those points (details below).

1. *on page 1, line 37, the authors claim that only the thermodynamic potentials and oxygen evolution overpotential define the overall potential required for water splitting. what about the hydrogen evolution reaction? Even with the best catalyst (Pt) there is some overpotential which should be accounted for.*

That is correct. We have rephrased this sentence of our introduction in a more accurate manner.

2. *Is there a reason the authors chose to fabricate the electrodes the way they did, with alumina templates and ALD post modification? This seems to be an overly complicated synthesis of iron oxide electrodes. This is certainly a large barrier for others to try and reproduce the results. If the results (as presented) are general, and not specific to the electrode preparation, then why not use a simpler system? Some comment about this in the start of the results section would be helpful.*

We have added the following explanatory sentence to the end of our introduction:

"These samples are chosen for their relatively high electrocatalytic activity (caused by the geometric surface area increase), which is necessary for the isotope analyses."

3. *figure 4 and related text. IR measurements have indicated a Fe=O intermediate species of water oxidation with iron oxide (the authors reference 16). Observation of an intermediate species must come before the rate determining step to be observed. This supports the authors conclusion of rate-determining O-O bond formation and is worth noting.*

Thank you for this interesting remark, which we have added to our discussion (line 269).

Reviewer #3

The publication „Direct oxygen isotope effect identifies the rate-determining step of electrocatalytic oxygen evolution reaction at a metal oxide surface“ by Haschke et al. describes a very interesting and new approach to investigate the oxidation of water at electrode materials. Apart from other in the article mentioned methods to investigate the involved processes, this new method provides a much deeper view on the involved processes. Especially, in the field of finding new electrode materials, this method will be useful. The publication is well structured and written in good English. Isotope data are well and clear presented and the statistical evaluation of the data is good. Due to the novelty and for discussion of the method in a broader scientific community, I recommend a publication in Nature Communications.

No corrections requested.

REVIEWERS' COMMENTS:

Reviewer #1 (Remarks to the Author):

The authors have made a good effort to address all of the points raised in the first review. I recommend that the revised manuscript should now be accepted.

Reviewer #2 (Remarks to the Author):

The revised manuscript by Haschke et al. adequately addressed reviewer comment and is suitable for publication in Nature Communications as is.